# Electronic recording of lifetime locomotory activity patterns of adult medflies

**Vasilis G. Rodovitis**[1], **Stella A. Papanastasiou**[1], **Evmorfia P. Bataka**[2], **Christos T. Nakas**[2,3], **Nikos A. Koulousis**[4], **James R. Carey**[5,6], **Nikos T. Papadopoulos**[1]*

**1** Laboratory of Entomology and Agricultural Zoology, Department of Agriculture, Crop Production and Rural Environment, University of Thessaly, Volos, Greece, **2** Laboratory of Biometry, Department of Agriculture, Crop Production and Rural Environment, University of Thessaly, Volos, Greece, **3** University Institute of Clinical Chemistry, Inselspital, Bern University Hospital, University of Bern, Bern, Switzerland, **4** Laboratory of Applied Zoology and Parasitology, School of Agriculture, Aristotle University of Thessaloniki, Thessaloniki, Greece, **5** Department of Entomology University of California Davis, Davis, California, United States of America, **6** Center for the Economics and Demography of Aging University of California Berkeley, Berkeley, California, United States of America

* nikopap@uth.gr

**Data Availability Statement:** Data is available from https://data.world/vasrod43/lam25system-medfly2021

## Abstract

Age-specific and diurnal patterns of locomotory activity, can be considered as biomarkers of aging in model organisms and vary across the lifetime of individuals. The Mediterranean fruit fly (medfly), *Ceratitis capitata*, is a commonly used model-species in studies regarding demography and aging. In the present study, we introduce a modification of the automated locomotory activity electronic device LAM25system (Locomotory Activity Monitor)–Trikinetics, commonly used in short time studies, to record the daily locomotory activity patterns of adult medflies throughout the life. Additionally, fecundity rates and survival of adult medflies were recorded. Male and female medflies were kept in the system tubes and had access to an agar-based gel diet, which provided water and nutrients. The locomotory activity was recorded at every minute by three monitors in the electronic device. The locomotory activity of females was higher than that of males across the different ages. For both sexes locomotory rates were high during the first 20 days of the adult life and decreased in older ages. The activity of males was high in the morning and late afternoon hours, while that of females was constantly high throughout the photophase. Negligible locomotory activity was recorded for both sexes during the nighttime. Males outlived females. Fecundity of females was higher in younger ages. Our results support the adoption of LAM25system in studies addressing aging of insects using medfly as a model organism.

## Introduction

Aging is the gradual decline in physiological integrity and function combined with increased chance of death and reduced probability to reproduce [1,2]. "Normal" aging is defined as the summary of physiological changes that occur in an organism over time [3], while "functional senescence" describes the progressive decline in activity and function with aging [4]. Age-related behavioral changes result in progressive deterioration of many spontaneous responses

**Funding:** Research funded in part by a grant from the Center for the Economics and Demography of Aging, UC Berkeley (NIH 2P30AG012839).

**Competing interests:** The authors have declared that no competing interests exist.

including locomotory activity [3]. The spontaneous behavior of organisms is directly associated with movement. It catalyzes the ability of the individual to respond to environmental stimuli and to daily vital needs, such us survival and reproduction, which are firmly correlated with its fitness traits and age [5].

Model organisms are non-human species which are used in biological research to investigate, among others, complex behaviors and biological phenomena and to provide information regarding life history traits of humans and mammals [6,7]. Fruit flies' activity and other specific behaviors (e.g. sexual signaling, supine behavior, feeding) vary during the life course of the individuals, with some of them expressing age related patterns. As a result, fruit flies can be exploited in demographic, aging and behavioral research [8–11]. Changes in spontaneous behavior and locomotory activity in relation to age had been described in *Drosophila* species and the Mediterranean fruit fly (medfly), *Ceratitis capitata* (Wiedemann) (Diptera: Tephritidae), providing insights to understand the aging process [8,11–15].

In this study, we used medfly a common model organism in demographic and gerontology research, to test age related locomotory activity [11,14,16–18]. Previous studies showed that the daily frequency of supine behavior was positively correlated with the time of the individual's death, while the frequency of performing sexual signaling by the males was negatively correlated with the predicted time to death [17,19,20].

Differences of daily behavioral patterns in adult medflies were previously attributed to sex and age. Generally, regardless the mating status, males are more active and longer-lived than females and sexual signaling represents their foremost activity during the day [21]. A voluminous part of literature examines the age-related patterns of Drosophilids' and Tephritids' behavior [3,4,11], but only few of them examine the details of locomotory activity throughout the lifespan of individuals, which is challenging to address (e.g. time- consuming and labor intensive). A large number of individuals is impossible to be observed visually. Though, the available commercial laboratory equipment is not easily applicable for lifetime studies yet, as such large datasets are difficult to be handled. Nevertheless, the combination of lifetime locomotory activity with survival and egg production is expected to generate important data towards understanding functional senescence in model organisms.

In order to study the age-related patterns of locomotory activity in medflies we used the LAM25system, an automated electronic device. Spontaneous activity and behavior of flies have been investigated in the past and ranged from simple visual observation to the use of various electronic devices [5]. The simplest approach is the "grid of square" method [22], where flies are kept individually in a chamber with a grid of circles drawn on its base. The number of crossed circles by the adult define the level of activity [22]. Ewing [23] developed the "labyrinth" method, where a series of cones allowed a unidirectional passage of individuals. Flies were released in the first cone-compartment and the distribution of them in the next compartments, generated data to assess the level of activity. Devices such as the "Actimeter" and the "Locomotron", consisting of a small chamber with an infrared light gate located in the center, are examples of automated equipment for measuring activity. An activity event is recorded whenever the fly crosses the light gate. Several variants of these apparatus have been developed, depending on the purpose of the study. Automated devices based on video-camera tracking systems were greatly improved in the last decade and were used to record the activity of fruit flies and *Drosophila* species [14,24–26]. For example, Zou et al. [14] and Chiu et al. [27], using a Behavioral Monitoring System- BMS, recorded the lifetime daily behavior in females of *Anastrepha ludens*, in relation to food quality, revealing different diurnal patterns of behavior of young, middle-aged, and old adults. The need for constant monitoring individual insects, under fluctuating laboratory conditions and for long time periods, led to the development of high quality, automated devices to track their movement [28]. The LAMsystem- Locomotory

Activity Monitor and DAMsystem- Drosophila Activity Monitor (Trikinetics), are extensively used devices for this type of assays [28–30].

The effect of different environmental and rearing conditions on the locomotory activity of fruit flies was traditionally measured by visual observations, which are time consuming, and observer biased. During the last two decades, the introduction, of the automatic DAM and LAM systems simplified the study of locomotory activity of fruit flies. For example, using the LAM system (in the frameworks of quality control tests of Sterile Insect Technique programs), the daily locomotory activity of *Bactrocera tryoni* adults was found to be positively correlated with protein and nutrient concentration in adult diet [26,31]. Utilizing the same system, Dominiak et al. [32] defined the effect of radiation inducing sterilization, on the activity of adult *B. tryoni*. Additionally Weldon et al. [24], reported the differences in daily behavior between fertile and sterile adults, by using a video system. Recently, the LAM system was used to determin differences in the activity patterns of *B. tryoni* adults with or without access to a diet rich in endosymbiotic bacteria [33], and of a wild-type and a self-limiting transgenic strain of *Bactrocera oleae* population [34].

Day and night lifetime locomotory activity has not been previously assessed in *C. capitata* adults. The aim of the present study was to establish a system to automatically record the activity of adult fruit flies throughout their lifespan as a foundation for research in aging and health dynamics using invertebrate model systems. For this purpose, we used, for the first time, the LAM25system by Trikinetics which records the movement of the flies on three different spots within the chamber where they were individually kept. Also, we recorded the fecundity of females with daily visual observations.

## 2. Materials and methods

### 2.1. Flies and rearing conditions

The study was conducted in the Laboratory of Entomology and Agricultural Zoology at the University of Thessaly, Greece, from September 2019 to December 2020, under constant laboratory conditions, $25 \pm 2°C$, $65 \pm 5\%$ RH, and a photoperiod of L14:D10 with photophase starting at 7:00 and ending at 21:00. Light intensity ranged from 1500 to 2000 Lux. We used males and females of the Mediterranean fruit fly laboratory strain "Benakio", that was maintained under laboratory conditions for more than 30 years [35].

The rearing procedure of flies was implemented as described in [36] and [37], by keeping adults in groups of 100 individuals in wooden, wire-screened cages ($30 \times 30 \times 30$ cm) provided, *ad libitum*, with water and an adult diet consisting of a mixture of yeast hydrolysate, sugar, and water at a 4:1:5 ratio. Females deposited eggs on the inner surface of an artificial substrate (dome), comprised by a hollow, red plastic hemisphere (5 cm Ø), which was punctured with 40–50 evenly distributed holes (1 mm Ø). Each dome was fitted in the cover of a plastic Petri dish (5.5-cm Ø). A plastic cup with 0.5 ml of orange juice was added in every Petri dish to stimulate oviposition. Water was placed in the Petri dish to maintain high humidity levels in the dome for female oviposition [38]. Domes were placed in rearing cages for 24h to collect eggs for the experimental needs. Collected eggs were placed on a cotton disk (5.5- cm Ø, 2 mm width), serving as a bulking agent for larval diet, within a petri dish. We placed 100 eggs per cotton disk which was previously soaked in the larval diet consisting of 200 g sugar, 200 g brewer's yeast, 100 g soybean flour, 4 g salt mixture, 16 g ascorbic acid, 16 g citric acid, 3 g sodium propionate, and 1 l water [38]. Petri dishes were placed into plastic containers on a layer of sterilized sand, where larvae pupated.

## 2.2. Assaying locomotory activity patterns of adult medflies

We recorded the locomotory activity patterns of adult medflies by using the Locomotory Activity Monitor- LAM25system (Trikinetics). In this system, flies were individually kept in 32 glass tubes (25 mm Ø, 125 mm length). Each tube was vertically crossed by three ray-rings of infrared light beams (monitors) at three different planes (close to the two ends of the tube and one in the middle) (Fig 1A). Activity was measured every minute, as the number of times the fly passed through an infrared beam.

The day that adults emerged, fifteen unmated males and sixteen unmated females were individually positioned in the tubes and stayed until death. Maintenance of flies in the tubes was ensured with minimum frequency of disturbances (jolts, flicks, shakes) that would affect their normal activity. We prepared an agar-based gel diet consisting of sugar, yeast hydrolysate, agar, nipagin and water (4:1:0.2:0.1:20) to provide adults with both nutrients and water simultaneously. The gel diet was replaced every four days to avoid dehydration.

On the one end of the tube, we adjusted a vinyl plastic stopper (CAP25-BLK- Vinyl Tube Cap- Trikinetics) with the gel diet and on the other end (Fig 1B), a cap with organdie cloth (Fig 1C), which served for the ventilation of tubes and also as an oviposition substrate (Fig 1D) [39]. Outside the oviposition substrate end of the tube, we adjusted a wooden, custom-made frame with four Plexiglas platforms (Fig 1E), where the eggs deposited by each female landed on a black filter paper. Adult mortality and the number of deposited eggs were daily recorded with visual observations until the death of all individuals.

## 2.3. Data processing and analyses

We calculated the mean daily (24h) locomotory activity of every male and female, recorded in one-minute intervals throughout their lifetime. The LAM25system was set to record, in each of the three monitors, the sum of movements an adult performed every minute. To calculate the mean activity of an individual for a given period of time, we summarized the recordings of the three monitors per minute during this period. Then, we summarized the recordings for every individual during 24h (1440 recordings), during daytime (840 recordings) and during nighttime (600 recordings). Accordingly, we constructed three event-history diagrams (24h interval, daytime– 14h interval, and night– 10h interval). An event-history diagram [40], depicts age-specific activity patterns using different color-coding for different levels of activity. Adult flies are represented in the diagram by horizontal "lines" proportional to their lifespan. Activity is color-grouped according to the summarized movements of an adult during the tested period. Three activity levels are represented by a separate color (low-green, medium-yellow, high-red).

Additionally, we calculated the mean activity per 24h interval for the cohorts of males and females, recorded during four selected ages (2, 11, 21, 42 and 81 days old). More specifically, for every individual, we summarized the recordings of the three monitors per minute, during the 24h interval for each selected day. The five selected ages were characterized as: newly emerged adults (2-days old), reproductively mature adults (11-days old), middle-aged adults (21-days old), elderly adults (42-days old) and extreme aged adults (81 days old) [6,41].

Survival analysis for males and females during their maintenance in the tubes of LAM25system was conducted with IBM SPSS 26.0 (IBM Corp., Armonk, NY, U.S.A.). Differences in survival between the two sexes were tested with the Cox proportional hazards model [37].

Activity data were analyzed with R v.4.0.0 (R Foundation for Statistical Computing, Vienna, Austria) using the "geepack" package for Generalized Estimating Equations (GEEs) [42] to account for within-subject dependency. Clusters of subjects that were considered within subject observations are correlated (time-dependency). The Poisson model was used for the

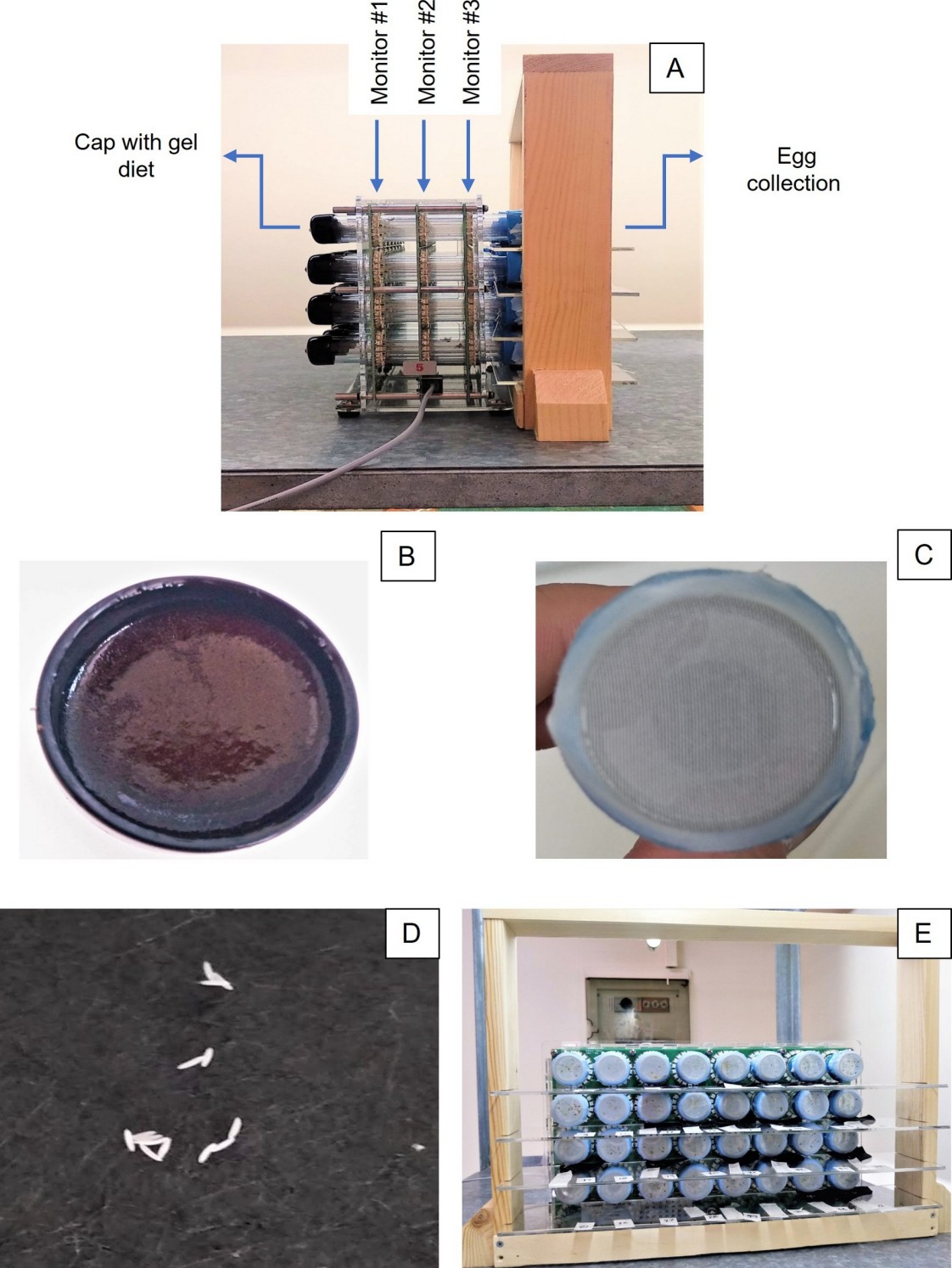

**Fig 1.** Side view of the LAM25system set in the laboratory (A). Vinyl plastic stopper placed on one end with the agar-based gel diet (B). Cap with organdie cloth placed on the other end for ventilation of tubes (C) and for egg deposition (D). Wooden- plastic, handmade construction, for egg collection (E).

response variable "Activity" since the response variable represents count data (sum activity/fly/day). In addition, we tested the effect of age and sex on average daily activity of flies. The average daily activity was estimated by the sum of daily movements recorded on the three monitors for each fly per minute, divided by the total number of flies which survived each day until the age of 91 days. Finally, the total activity of flies per time period (night, 21:00–6:59; morning, 7:00–13:59; midday, 14:00–17:59; evening, 18:00–20:59) was compared between males and females for 5 different ages (2, 11, 21, 42, 81 days old). Parameter estimates are presented as Incidence Rate Ratios (IRR) with 95% confidence intervals (CI), which is the ratio of the activity level in a group of interest to the number of activities of the group used as reference. To interpret the results, for IRR >1, the activity is higher for the group of interest rather than the reference group, while IRRs lower than 1 indicate less activity for the group of interest comparing to the reference. Pairwise contrasts for estimated marginal means were assessed using the Tukey adjustment for multiple comparisons.

## 3.Results

### 3.1. Demographic parameters

Males lived longer than females (Wald $\chi^2$ = 5.38, df = 1, $P$ = 0.02) (Fig 2). The first dead adult was observed on day 48 and on day 53 of age, for males and females, respectively. Female mortality rates increased dramatically after the age of 50 days compared to males. Half of the female and male cohorts survived 59 and 74 days, respectively (Fig 2A). The lifespan of the longest-lived male (166 days) differed by 75 days from that of the longest-lived female (91 days).

The mean lifetime fecundity was 973.4 ± 56.6 (SE) eggs per unmated female. Females reached their maximum oviposition rate on the 13th day of age with an average of 42.6 ± 4.6 eggs (Fig 3). Fecundity rates followed a linear downward trend from day 30 of age onwards.

### 3.2. Activity

**Age-specific locomotory activity.** The age-specific locomotory activity patterns of males and females is presented in Fig 4. Activity levels were estimated for the whole 24h interval, and for daytime and nighttime separately. Females were more active than males during the 24h and the daytime periods regardless of their age (Table 1; Fig 4). The locomotion activity of both sexes declined with age and was not affected by the tested period (24h, daytime, nighttime). No significant differences were recorded in activity levels between the two sexes during nighttime (Table 1). The pattern of declining activity with aging of males both during day- and nighttime was smooth and progressive. Age-specific patterns in female activity were high during the first 20 days of adult life, declining sharply to day 30 stabilized up to approximately day 45 and then declining again to day 60 and stabilized to day 80 of age. Age-specific patterns of females differed a great deal between day and night.

An event history chart was used to depict the detail of age-specific locomotory activity of males and females for the whole 24h period, as well as for daytime and nighttime periods (Fig 5). The females were found to be more active than males throughout their lifespan. Males exhibit high levels of locomotory activity (>2000 movements) during the first 10 days of their life. Activity levels remained at a rather medium level (501–2000) from day 10 to day 60 and further reduced in older ages. The activity of females was high for all individuals during the first 20 days. High variability in activity patterns among females was reported from day 30 to 60. During nighttime, no different patterns in age-specific locomotory activity were observed between the two sexes. All flies were moribund for one or two days before death.

**Fine-scale 24h locomotory activity rhythms.** Both males and females were more active during daytime (from the beginning until the end of the photophase) and remained almost

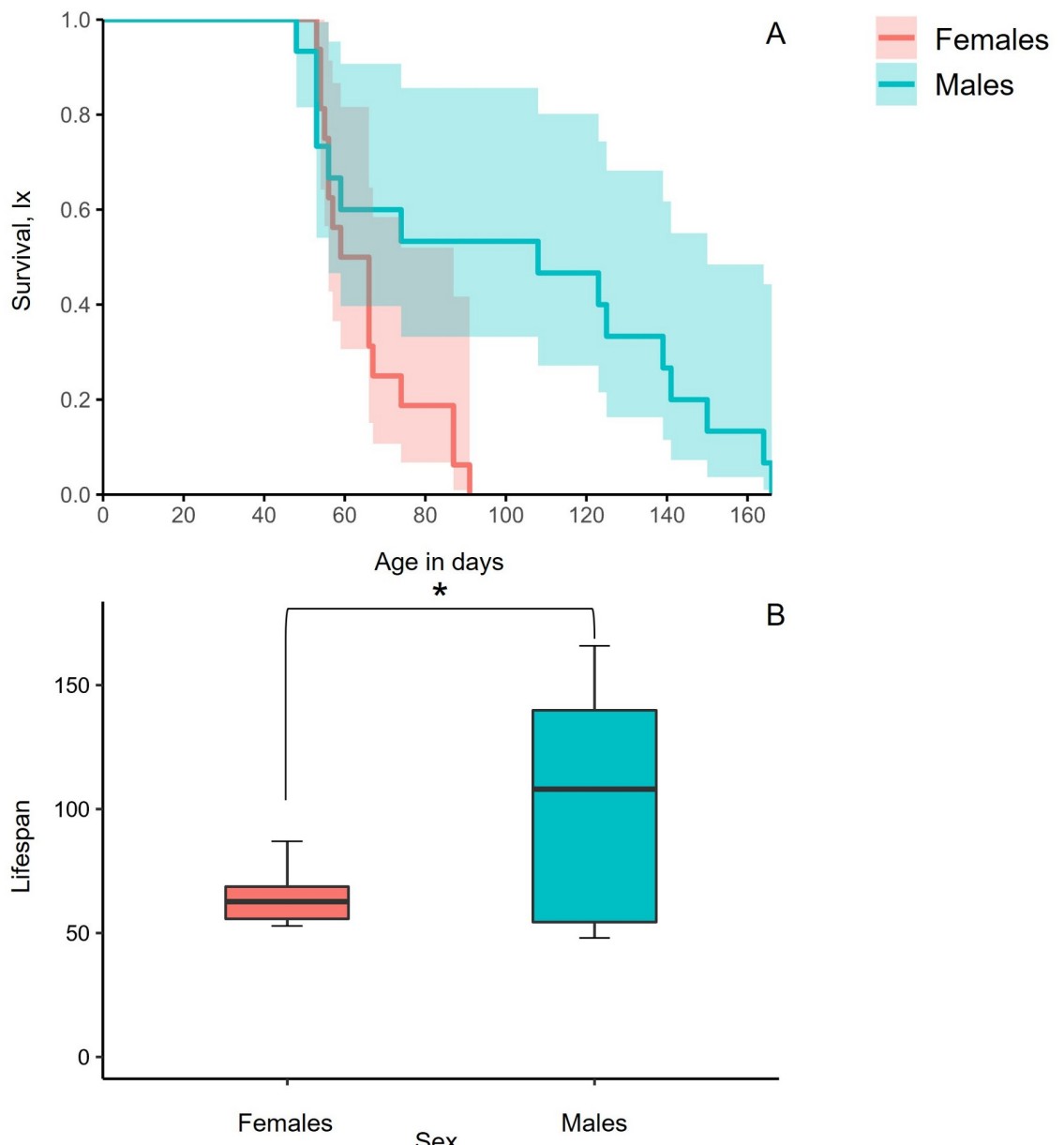

**Fig 2.** Age-specific survival curves with 95% Confidence Intervals (CI) (A) and box plots depicting lifespan of males and females (B) kept in the tubes of the LAM25system.

motionless during darkness, regardless of the age group (Fig 6). Locomotory activity started at approximately 06:00, shortly before the start of the photophase (7:00) and stopped at approximately 22:00 shortly after its end (21:00), lasting for almost 16 hours. Flies of the different age groups exhibited different patterns of locomotory activity. Therefore, we separated the 24h interval in four different periods (night, morning, midday and evening) to compare males and females (S1 Table). Male medflies exhibit a specific diurnal pattern of sexual signaling characterized by increased intensity in the morning and afternoon hours as it was demonstrated earlier [8,19,36].

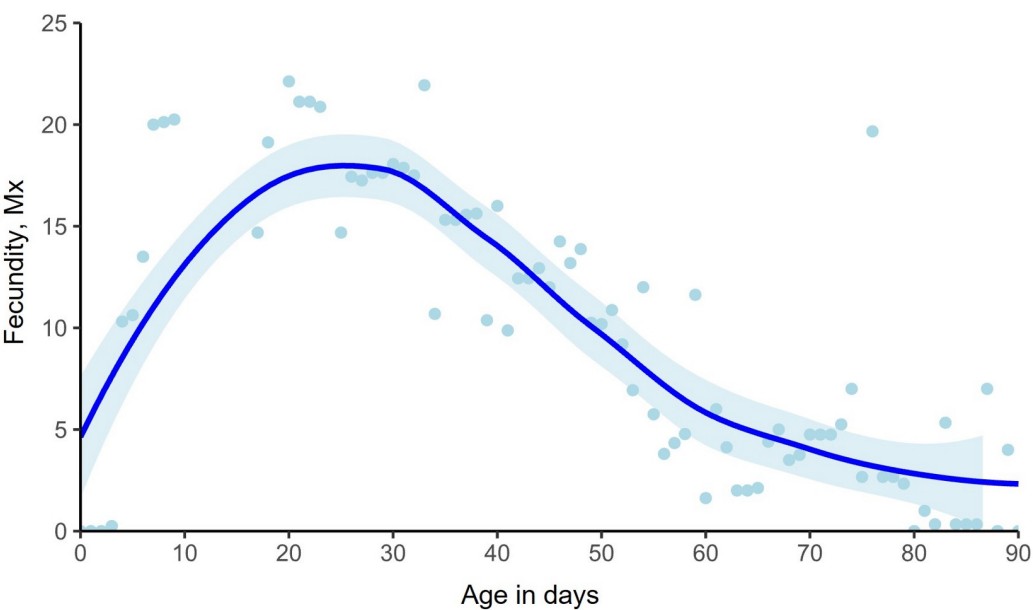

**Fig 3. LOESS curves (Locally Estimated Scatterplot Smoothing curves) for age-specific fecundity rates with 95% Confidence Intervals (CI) of females kept in the tubes of LAM25system.**

No significant differences were recorded in the activity patterns between males and females on the 2nd and 81st day of life, adjusting for day period (S1 Table). During the photophase, flies were more active in the morning and less active in the midday and the evening. Reproductively mature (11-days old), middle-aged (21-days old) and elderly (42-days old) females were more active than males, adjusting for day period. Flies of the three respective ages were more active during the morning than in the midday and evening. The lowest activity was recorded during the night for all five ages tested (S1 Table).

## 4. Discussion

This is the first study on the fine-scale patterns of lifetime locomotory activity for adult medflies. A major advance of our study includes the adaptation of the LAM25system for long-term activity recordings (almost six months), as until now this system was used only for short time studies (few days), such us recording activity levels of newly emerged adult fruit flies, mainly addressing quality control issues, for their use in SIT programs [24,26,31,32]. Also, the same system was implemented to characterize the circadian rhythms and sleep patterns of other Diptera species such as Drosophilids but again only for a few days [28,29,43]. However, Koh and co-workers [12], measured lifetime activity of *D. melanogaster* adults using the DAMsystem, but flies had to be transferred often to new vials to ensure access to fresh food. Apart from extending the study of locomotory activity to the whole lifespan of the tested organism, we managed to simultaneously record the lifetime fecundity of females. For this purpose, we developed and adjusted an egg laying system to the LAM25system, while flies were kept in good health in the tubes. Our approach enabled the recording of adult survival and female fecundity daily, without causing disturbances in the system that would affect the activity of the flies. The current modified system can be used to answer questions regarding (a) the relationships among reproductive effort (i.e., fecundity), activity patterns and female longevity, (b) the aging process, and (c) health dynamics using medfly as a model organism.

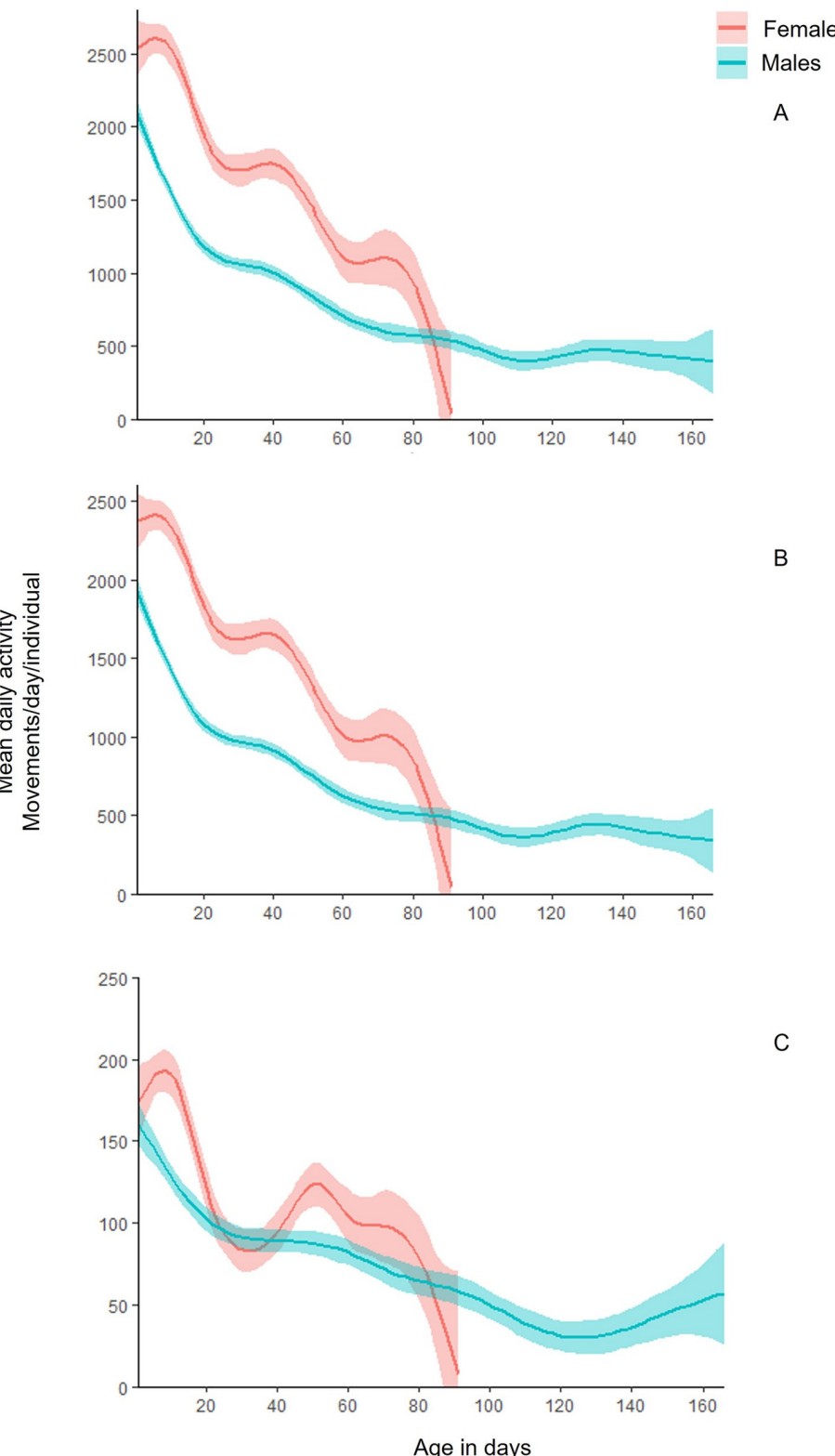

**Fig 4.** LOESS curves (Locally Estimated Scatterplot Smoothing curves) for mean activity of males (blue) and females (red) with 95% Confidence Intervals (CI), during the 24h period (A), daytime (B) and nighttime (C). Mean activity was estimated by the sum of movements of all individuals during the period of interest (24h, daytime, nighttime), divided by the number of individuals surviving on each day.

**Table 1. Variables of the GEEs with significant effects on the age-specific activity of flies during the 24h day-night interval, the daytime and the nighttime periods, for their entire lifespan, by the factors: Sex (males, females) and age.**

| Factor | Day and night | | Day | | Night | |
|---|---|---|---|---|---|---|
| | IRR (95% CI) | P value | IRR (95% CI) | P value | IRR (95% CI) | P value |
| **Sex** (ref: female) | 0.710 (0.623, 0.809) | < 0.001 * | 0.687 (0.605, 0.782) | < 0,001 * | 0.858 (0.698, 1.050) | 0.144 |
| **Age** | 0.986 (0.983, 0.988) | < 0.001 * | 0.985 (0.983, 0.987) | < 0,001 * | 0.989 (0.987, 0.992) | < 0,001 * |

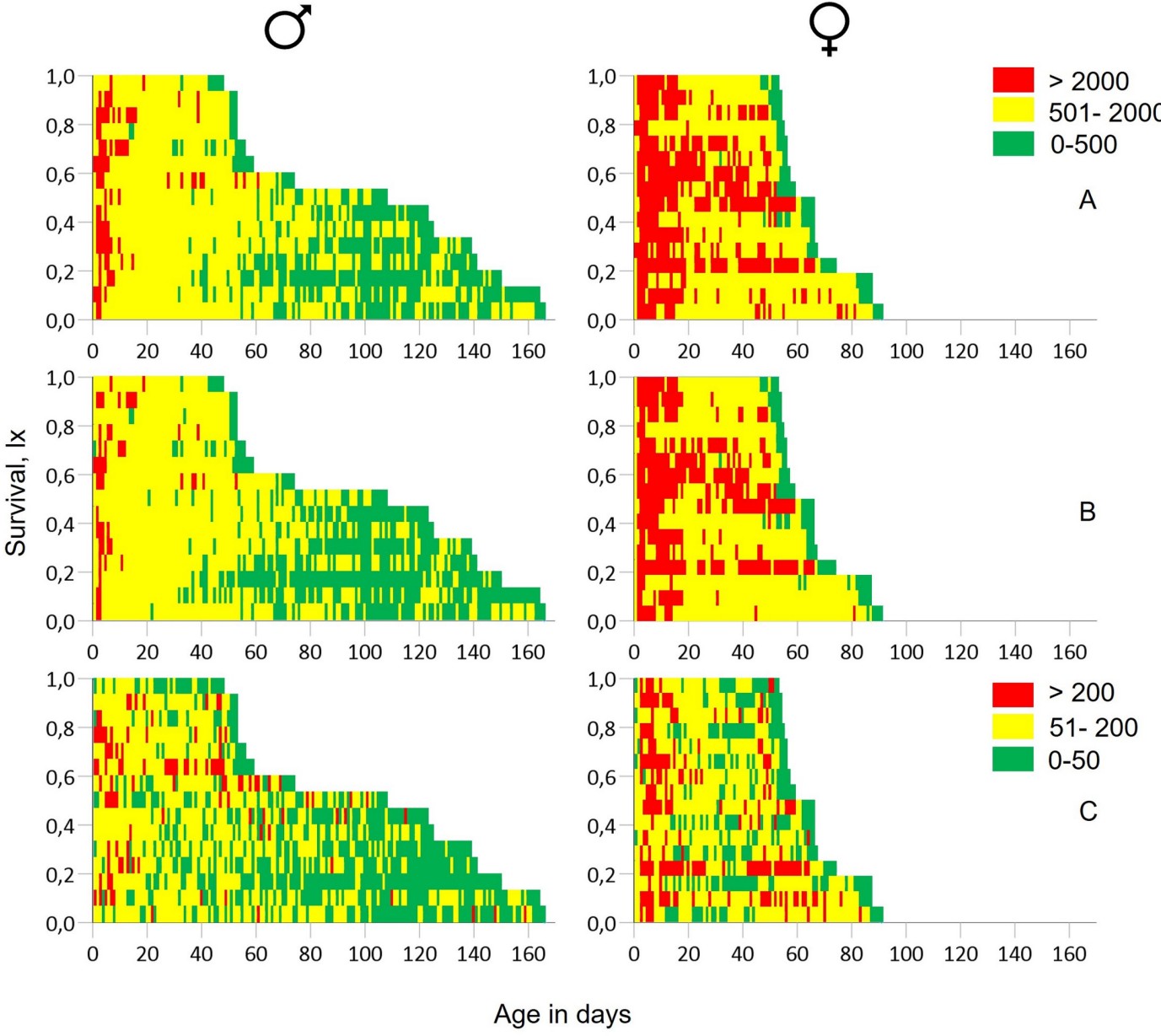

**Fig 5.** Event history diagram providing a detailed picture of the ages when activity was recorded during the 24h interval (A), daytime (B) and nighttime (C). Red: > 2000 movements during 24h and daytime and > 200 during nighttime, yellow: 501–2000 movements during 24h and daytime and 51–200 during nighttime, green: 0–500 movements during 24h and daytime and 0–50 during nighttime.

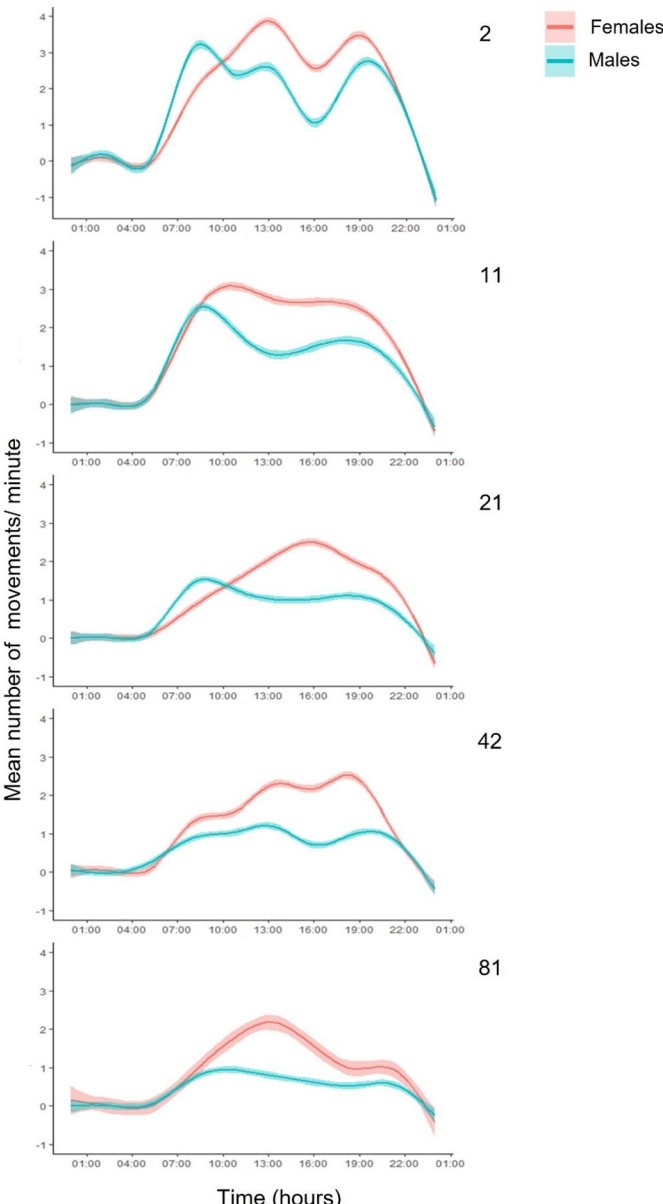

**Fig 6. LOESS curves (Locally Estimated Scatterplot Smoothing curves) with 95% Confidence intervals (CI) of the mean 24h locomotory activity for males and females at ages 2, 11, 21, 42 and 81 days.**

In agreement with previous studies, our results demonstrate that males outlived females [44–47]. Females face higher reproductive cost due to the reproductive system attrition caused by egg production and oviposition events and this cost leads on decreased longevity. Females medflies exhausting their reproductive potential rapidly, are predicted to have shorter lifespan [41]. However, in males the higher rates of sexual signaling in young ages is strongly correlated with longer lifespan [48]. Males face a lower cost of reproduction than females [8,21]. In our study, young females (9th-17th day) laid more eggs, while from the 35th day onwards the fecundity rates substantially decreased, and the lowest egg production was recorded in older

females. According to the models of Müller et al. [41], and Novoseltsev et al. [49], after attaining peak rates the fecundity of female medflies decreases rapidly, and the mortality gradually supervenes. Our results are consistent with both studies above since mortality rates of females increased rapidly after day 50 and daily egg laying dramatically deteriorated after peak was recorded. Young female medflies invest high levels of energy in egg production and older ones in survival [18], but this strategy is not sufficient to prolong lifespan in comparable levels to that of males.

Virgin females and non-mated males were considered in our current study and flies were kept in isolation. Therefore, effects of mating *per se*, reproduction related social interactions and effects of crowding that may negatively affect both health and longevity of adult medflies were not considered [50–52]. In preliminary trials, we found longer longevity and higher reproductive rates of female medflies kept at the LAM tubes than in our standard, plastic individual, demographic cages (S2 Table and S1 Fig). Restriction of flight may save energy reserves that can be invested to reproduction and soma maintenance extending life span. Similar conclusions were obtained by Koveos et al., [53], when the olive fly adults (*Bactrocera oleae*) were kept in small cages that did not allow flight.

Age-specific locomotory activity showed a declining trend with age in both sexes. Females were more active than males throughout lifespan, in contrast to the expression of other specific behaviors of medfly [21]. In other tephritid species, such as *B. tryoni* [31] males were more active than females. Medfly males were found to perform sexual signaling with greater frequency and duration throughout most of the photoperiod [8,19,21]. Apparently, males performing sexual signaling remain stationary and hence no activity/ movement can be recorded by the LAM25system. Sexual signaling is an indicator of longevity, and higher frequency is positively correlated with males' survival [19,21,54,55]. The LAM25system can only detect movement and therefore could not record the sexual signaling behavior of males. Although not directly measured, we can assume that differences in age-specific locomotory activity between males and females resulted mainly because males invested most of the time during photophase in performing sexual signaling (8, 19, 20, 21).

Locomotory and fecundity rates of female medflies were high at young ages (first 20 days) and then gradually declined. A progressive decrease in daily activity with increasing age was previously reported in female medflies [11,17,19,21], and in both sexes of *A. ludens* [14] and of *D. melanogaster* [3,4,56]. Walking, flying, and feeding frequencies were higher at young ages of medfly and *D. melanogaster*, in contrast to other motionless behaviors like resting and grooming, which remained in similar levels despite aging [11]. Consistent with our results, daily activity patterns of *Drosophila* spp. differed by sex, age and reproductive maturity [5,12,57] with young flies being more active and fecund than older ones. According to Bochicchio et al. [57], and references therein, aging is directly associated with the gradual reduction of activity and the expression of activity is an indicator of the "functional state" of the individual. Specifically, aging implies a progressive deterioration in the ability to maintain homeostasis, metabolism, and normal functions at high levels [58]. Research concerning the functional deterioration in relation to age identifies key organic systems that fail to function satisfactory in older ages [59]. Both the modification of LAM25system to record lifetime locomotory activity and the results of our study could assist towards understanding the aging process in insect model organisms.

Consistent with previous studies on activity of Tephritidae, our results showed that medfly locomotory activity patterns of selected ages (2, 11, 21, 42, 81 days) were higher during the photoperiod [51]. Generally, activity levels differed between male and females [8,36,54,60–65]. Specifically, both sexes were more active in the morning than in midday and evening. Visual observations confirmed that during evening both sexes invested time in the less active behavior

of feeding [54,60,61]. During early morning period (7:00–10:00), males exhibited similar activity with females, but females were more active throughout the rest of the day. Previous studies also showed that early in the morning, males in nature are involved in lekking activities trying to establish or locate leks [54,60,61]. Conversely, they exhibit mainly sexual signaling and remain stationary during the morning and midday [8,36,60]. A slight increase of activity observed during late evening (on 7:00 pm) could be explained by male re-activation and exhibition of sexual signaling as described earlier by Papadopoulos et al. [8] and Diamantidis et al. [36]. According to Warburg & Yuval [54] and Díaz-Fleischer & Aluja [61], female tephritids perform all type of behaviors throughout the day, without these behaviors being separated by the time period of day, which is consistent with our findings.

## 5.Conclusion

Our study highlights the modification and use of the LAM25system to record individual lifetime locomotory activity patterns of true fruit flies, in combination with fecundity and survival. We revealed a strong connection between decreasing locomotory activity and fecundity with aging. This system could be exploited in studies of demography and aging process, provided that these factors are strongly correlated with functional senescence of true fruit flies. In addition, our experimental procedure could be used in quality control tests performed in SIT, or in studies dealing with stress physiology and phenotypic plasticity.

Although, the trials reported in the current paper have been conducted with a limited number of individuals, we successfully demonstrated several important patterns of activity in medflies, differences between males and females and a connection of activity with aging and fecundity. Hence, our study forms the basis for future exploration of the correlation among locomotory activity, functional aging, and reproduction elements.

## Supporting information

**S1 Fig. Age-specific survival curves with and box plots depicting lifespan of males and females kept in the tubes of the LAM25system and in the plastic cup cages.**
(PNG)

**S1 Table. Variables of the GEEs with significant effects on the activity of flies by the factors: Sex (males, females) and time period (night, morning, midday, evening) of days 2, 11, 21, 42, 81.**
(DOCX)

**S2 Table. Average fecundity and duration of reproduction periods in days (pre- oviposition, oviposition, and post- oviposition) of females kept in tubes of LAM25system and in standard plastic cup cages.**
(DOCX)

## Author Contributions

**Conceptualization:** Vasilis G. Rodovitis, Stella A. Papanastasiou, Nikos A. Koulousis, James R. Carey, Nikos T. Papadopoulos.

**Data curation:** Vasilis G. Rodovitis, Stella A. Papanastasiou, Evmorfia P. Bataka.

**Formal analysis:** Vasilis G. Rodovitis, Evmorfia P. Bataka, Christos T. Nakas, Nikos T. Papadopoulos.

**Funding acquisition:** James R. Carey.

**Investigation:** Vasilis G. Rodovitis.

**Methodology:** Vasilis G. Rodovitis, Stella A. Papanastasiou, James R. Carey, Nikos T. Papadopoulos.

**Project administration:** James R. Carey, Nikos T. Papadopoulos.

**Resources:** James R. Carey, Nikos T. Papadopoulos.

**Supervision:** James R. Carey, Nikos T. Papadopoulos.

**Writing – original draft:** Vasilis G. Rodovitis, Stella A. Papanastasiou, Nikos T. Papadopoulos.

**Writing – review & editing:** Vasilis G. Rodovitis, Stella A. Papanastasiou, Evmorfia P. Bataka, Christos T. Nakas, Nikos A. Koulousis, James R. Carey, Nikos T. Papadopoulos.

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
