## [Decision Letter · Decision Letter 0]

14 Feb 2022

PONE-D-22-00246Electronic recording of lifetime locomotory activity patterns of adult medfliesPLOS ONE

Dear Dr. Papadopoulos,

Thank you for submitting your manuscript to PLOS ONE. After careful consideration, we feel that it has merit but does not fully meet PLOS ONE’s publication criteria as it currently stands. Therefore, we invite you to submit a revised version of the manuscript that addresses the points raised during the review process.

Good job on this project and manuscript! Please review carefully the reviewer comments, these are opportunities to make a good paper even better.

We look forward to receiving your revised manuscript.

Kind regards,

Nicholas C. Manoukis

Academic Editor

PLOS ONE

Journal Requirements:

Reviewers' comments:

Reviewer's Responses to Questions

**Comments to the Author**

1. Is the manuscript technically sound, and do the data support the conclusions?

Reviewer #1: Yes

Reviewer #2: Yes

Reviewer #3: Yes

2. Has the statistical analysis been performed appropriately and rigorously? 

Reviewer #1: Yes

Reviewer #2: Yes

Reviewer #3: Yes

3. Have the authors made all data underlying the findings in their manuscript fully available?

Reviewer #1: Yes

Reviewer #2: Yes

Reviewer #3: Yes

4. Is the manuscript presented in an intelligible fashion and written in standard English?

Reviewer #1: Yes

Reviewer #2: Yes

Reviewer #3: Yes

5. Review Comments to the Author

Reviewer #1: The manuscript by Rodovitis et al. describes individual lifetime locomotor activity patterns in Ceratitis capitata (Mediterranean fruit fly) in combination with survival and female egg production. The manuscript is generally well written, and the experimental design is mostly sound. One concern I have with this manuscript is the use of virgin females used in this experiment to measure reproduction. Gravid females would likely produce more eggs at an earlier interval during the experimental period, which could potentially impact their survival. Additionally, the use of gravid females would permit the quantification of egg hatch, which is a measure of reproduction. The authors should provide a rationale for using virgin female flies as a proxy to measure fertility. In my opinion, this experiment measured the fecundity of virgin females, not reproduction. Changes should be made throughout the manuscript to correct this or additionally experiments should be performed that more accurately measure reproduction.

This manuscript should be considered for publication because C. capitata is a major agricultural pest and the monitoring system developed for these experiments will prove useful for other pestiferous Tephritidae species. My recommendation is a minor revision to address the concerns raised above and to consider the specific comments described below:

1) Lines 142 and 399: 7:00 am, 9:00 pm, and 7:00 pm are used. Please be consistent with time and use the 24-hour clock throughout the manuscript for time descriptions.

2) Line 211: change middleaged to middle-aged

3) Line 225-228: Move time descriptions to the beginning of the section

4) Line 317: change middleaged to middle-aged

5) Line 351: remove apparently, seems unnecessary.

6) Line 370-372: “Although not directly measured, we can assume that differences in age specific locomotory activity between males and females resulted mainly because males invested most of the time during photophase in performing sexual signaling.” Please provide references supporting this assumption.

7) Line 339: change reproduction to fecundity

8) Line: 412: change reproduction to fecundity

9) Line 433: double-check format for reference 3

10) Change age specific to age-specific throughout

Reviewer #2: Review of PONE-D-22-00246

Electronic recording of lifetime locomotory activity patterns of adult medflies.

General comments

Generally, the paper is well written and the subject matter is an interesting contribution to science.

Regarding the use of “medflies”, this is an abbreviation of “Mediterranean fruit fly”. Given that “Mediterranean” is a noun, it is capitalised. There is some logic that “medfly” should be “Medfly” if you follow the naming convention. We don’t write about the mediterranean sea – these are names and are capitalised.

There are not many suggestions, other than the abstract looks like it was written last, as it should be. Most edits are in the abstract, probably because this was the least text reviewed by the authors.

Writers should write for the reader. Therefore, words such as “also” and “therefore” should be at the start of the sentence to tell the reader that something additional is coming. Placing these words in the middle of the text does not prepare the reader for change.

I am not a fan of long caveats in front of factual statements. Say the important stuff first, then provide the circumstances (the caveat).

Your results “are consistent with” the results of other scientists. But technically, they are not in agreement. They might reach similar conclusions but they do not agree with each other. This “agreement” phrase needs to be replaced with the “consistent with” phrase.

I am not a fan of “has been” verbs, particularly when there are references that report things “were” or “was” found. “Has been” is less direct and less confident. They are not wrong but they are not direct.

Specific comments

Line 37- try “… activity can be considered … organisms and vary …”

Line 43 – try “Additionally, fecundity rates …” and delete the “as well”

Line 45 – try “… which provided and nutrients.”

Line 46 – try “… three monitors in the electronic …”

Line 50 – try “… that of females was constantly high …”

Line 64-67 – sentence too long – I aim at 28-35 words per sentence – try two sentences

Line 70-73 – sentence too long, try two sentences

Line 74 – this is the first use of “Mediterranean fruit fly” – the full name needs to be here and not in line 76.

Line 84 – try “… and age. Generally, males are …”

Line 91-92 – say the important stuff first. Try “Spontaneous activity … past and ranged from …”

Line 103 – try “… systems were greatly …” – you have references to support the statement

Line 114 – try “During the last two decades, the introduction of the …”

Line 118-121 – too long. Try a new sentence starting “Additionally, Weldon et al used an …”

Line 121 – try “Recently, the LAM system was …”

Line 131 – try “Also, we recorded …’

Line 172 – try “… tubes and stayed until death.”

Line 206 – try “Additionally, we calculated …”

Line 216 – I think it needs a reference

Line 222 – try “Additionally, we test …”

Line 239-244 – great text – very clear – well written

Line 250-252 – again, good clear text – easy to read and understand

Line 263 – I am not a fan of “as well” – try “and”. Why use two (or three) words when one word will do.

Line 293 – try “… females were found from …”

Line 308 – try “… 16 hours. Flies of the … exhibited …”

Line 309 – try “… activity. Therefore, we separated …”

Line 331 and 333 – replace “has been” with “was” – this is more direct and confident language

Line 347-349 – try “Females face higher reproductive cost due to … events and this cost leads to decreased longevity.”

Line 351 – technically, lifespan should be “longer” because it related to time. The number might be “higher” but lifespan is longer.

Line 356 – technically, we say “our results are consistent with …”. Alignment is something different.

Line 358 – do you mean “deteriorate” or “decorate”?

Line 358-360 – try “… recorded. Young female … survival [18] but this strategy…”

Line 362 – I think you need to state that more energy use leads to death or shorter lifespan

Line 368 – try “… and higher frequency is positively …”

Line 375 – replace “has been” with “was” – replace “as well as” with “and”

Line 378-379 – try “Consistent with our results, daily …”

Line 389 – try the “consistent with” phrase rather than “agreement”.

Line 389-392 – is very long – try two sentences.

Line 396 – try “… day. Also, previous studies reported …”

Line 397 – try “…59]. Conversely, they exhibit …during later in the morning and midday.”

Line 403 – use the “consistent with” phrase – not “agreement”

Line 411 – try “We found a …”. Technically, the results are a thing and it can’t do anything. Results can’t see or find or think. That is what the scientists, the humans, do. So use the “we” words. Results are something you create but results don’t do the thinking or showing.

Line 412-414. Say the important stuff first. Try “This system could be … aging process, provided that …”

Line 417-420 – sentence too long. Try “… the trials … individuals. We successfully demonstrated several …”

References – some are abbreviated – some are in full. Pick the journal standard and stick to it.

Reviewer #3: A well designed study investigating the lifelong locomotor activity of Medflies. Findings are providing valuable insights in the fecundity and activity patters of the flies during a period of many months (usually those studies are restricted in few 24 hrs periods for the purpose of evaluating quality of SIT flies. I would recommend this paper to be published, pending few minor corrections and suggestions that I submit for the consideration of the authors in the annotated PDF.

General comments:

1) It is known that social interactions and caging density affect the health of the flies, I would like to see some comment in discussion, since the experimental flies spent all their life under severely restricted solitary conditions.

2) The authors do mention the benefits of the modified LAM system for studying the activity patterns over very long periods of time. I would like to see in the discussion a brief comparison with other methods aiming to also access the orgamism's physiological status over long period of time (lipids-proteins from subsamples, video recording?)

3) It would be interesting to access the effects of mating status, by monitoring mated vs virgin females (if not in this study, it would be worthy to mention it as basis for future experiments?)

6. PLOS authors have the option to publish the peer review history of their article (what does this mean?). If published, this will include your full peer review and any attached files.

Reviewer #1: No

Reviewer #2: **Yes: **Bernie Dominiak

Reviewer #3: **Yes: **Polychronis Rempoulakis

---

## [Author Response · Author response to Decision Letter 0]

14 Apr 2022

A list of responses to each of the Reviewers’ specific comments is given bellow: 

Reviewer #1

The manuscript by Rodovitis et al. describes individual lifetime locomotor activity patterns in Ceratitis capitata (Mediterranean fruit fly) in combination with survival and female egg production. The manuscript is generally well written, and the experimental design is mostly sound. One concern I have with this manuscript is the use of virgin females used in this experiment to measure reproduction. Gravid females would likely produce more eggs at an earlier interval during the experimental period, which could potentially impact their survival. Additionally, the use of gravid females would permit the quantification of egg hatch, which is a measure of reproduction. The authors should provide a rationale for using virgin female flies as a proxy to measure fertility. In my opinion, this experiment measured the fecundity of virgin females, not reproduction. Changes should be made throughout the manuscript to correct this or additionally experiments should be performed that more accurately measure reproduction. This manuscript should be considered for publication because C. capitata is a major agricultural pest and the monitoring system developed for these experiments will prove useful for other pestiferous Tephritidae species.

Response: We definitely appreciate this comment of Reviewer #1. Indeed, referring to reproduction might not be appropriate and hence we replace the term reproduction by fecundity/egg production. Using gravid females and measuring hatch rates would be a very interesting subject for a future study. Measuring fecundity patterns of virgin female medflies is a common experimental method in demographic research of Tephritids. Whether mated, and virgin females exhibit different lifetime activity pattern would be an interesting topic of our future research.

Specific comments:

My recommendation is a minor revision to address the concerns raised above and to consider the specific comments described below:

1. Lines 142 and 399: 7:00 am, 9:00 pm, and 7:00 pm are used. Please be consistent with time and use the 24-hour clock throughout the manuscript for time descriptions.

Response: The time periods reported using the 24-hour clock throughout the text.

2. Line 211, 317: change middleaged to middle-aged 

Response: the word “middleaged” was replaced by “middle-aged”

3. Line 225-228: Move time descriptions to the beginning of the section.

Response: the text was revised and the time descriptions moved in the beginning of the section.

4. Line 351: remove apparently, seems unnecessary.

Response: removed

5. Line 370-372: “Although not directly measured, we can assume that differences in age specific locomotory activity between males and females resulted mainly because males invested most of the time during photophase in performing sexual signaling.” Please provide references supporting this assumption.

Response: the requested references added in the text

6. Line 339, 412: change reproduction to fecundity

Response: the term ‘reproduction’ was replaced throughout the text with the terms ‘fecundity’ and ‘egg production’

7. Line 433: double-check format for reference 

Response: the references were checked and reviewed throughout the text

8. Change age specific to age-specific throughout

Response: the term ‘age specific’ replaced by the term ‘age-specific’ throughout the text

Reviewer #2

Generally, the paper is well written and the subject matter is an interesting contribution to science. Regarding the use of “medflies”, this is an abbreviation of “Mediterranean fruit fly”. Given that “Mediterranean” is a noun, it is capitalised. There is some logic that “medfly” should be “Medfly” if you follow the naming convention. We don’t write about the mediterranean sea – these are names and are capitalised. There are not many suggestions, other than the abstract looks like it was written last, as it should be. Most edits are in the abstract, probably because this was the least text reviewed by the authors. Writers should write for the reader. Therefore, words such as “also” and “therefore” should be at the start of the sentence to tell the reader that something additional is coming. Placing these words in the middle of the text does not prepare the reader for change. I am not a fan of long caveats in front of factual statements. Say the important stuff first, then provide the circumstances (the caveat). Your results “are consistent with” the results of other scientists. But technically, they are not in agreement. They might reach similar conclusions, but they do not agree with each other. This “agreement” phrase needs to be replaced with the “consistent with” phrase. I am not a fan of “has been” verbs, particularly when there are references that report things “were” or “was” found. “Has been” is less direct and less confident. They are not wrong, but they are not direct.

Response: We appreciate the comment of the Reviewer #2 regarding writing of medfly – Medfly. We consider medfly as a common name similar to “dog” that’s why we do not capitalize the first letter. However, we could follow the suggestion of the reviewer if the editor thinks so. We thank the reviewer for the editing on the abstract that improves the text.

Specific comments:

1. Line 37- try “… activity can be considered … organisms and vary …”

Response: The text was revised according to reviewer’s suggestions 

2. Line 43 – try “Additionally, fecundity rates …” and delete the “as well” 

Response: The text was revised according to reviewer’s suggestions 

3. Line 45 – try “… which provided and nutrients.” 

Response: The text was revised according to reviewer’s suggestions 

4. Line 46 – try “… three monitors in the electronic …” 

Response: The text was revised according to reviewer’s suggestions 

5. Line 50 – try “… that of females was constantly high …” 

Response: The text was revised according to reviewer’s suggestions 

6. Line 64-67 – sentence too long – I aim at 28-35 words per sentence – try two sentences 

Response: The text was revised according to reviewer’s suggestions 

7. Line 70-73 – sentence too long, try two sentences 

Response: The text was revised according to reviewer’s suggestions 

8. Line 74 – this is the first use of “Mediterranean fruit fly” – the full name needs to be here and not in line 76. 

Response: The text was revised according to reviewer’s suggestions 

9. Line 84 – try “… and age. Generally, males are …” 

Response: The text was revised according to reviewer’s suggestions 

10. Line 91-92 – say the important stuff first. Try “Spontaneous activity … past and ranged from …” 

Response: The text was revised according to reviewer’s suggestions 

11. Line 103 – try “… systems were greatly …” – you have references to support the statement 

Response: The text was revised according to reviewer’s suggestions 

12. Line 114 – try “During the last two decades, the introduction of the …” 

Response: The text was revised according to reviewer’s suggestions 

13. Line 118-121 – too long. Try a new sentence starting “Additionally, Weldon et al used an …” 

Response: The text was revised according to reviewer’s suggestions

14. Line 121 – try “Recently, the LAM system was …” 

Response: The text was revised according to reviewer’s suggestions

15. Line 131 – try “Also, we recorded …’ 

Response: The text was revised according to reviewer’s suggestions

16. Line 172 – try “… tubes and stayed until death.” 

Response: The text was revised according to reviewer’s suggestions

17. Line 206 – try “Additionally, we calculated …” 

Response: The text was revised according to reviewer’s suggestions

18. Line 216 – I think it needs a reference 

Response: A reference was added according to re reviewer’s suggestion 

19. Line 222 – try “Additionally, we test …” 

Response: The text was revised according to reviewer’s suggestions

20. Line 239-244 – great text – very clear – well written thanks bro

Response: The text was revised according to reviewer’s suggestions

21. Line 250-252 – again, good clear text – easy to read and understand thanks bro

Response: The text was revised according to reviewer’s suggestions

22. Line 263 – I am not a fan of “as well” – try “and”. Why use two (or three) words when one word will do.

Response: The text was revised according to reviewer’s suggestions

23. Line 293 – try “… females were found from …” 

Response: The text was revised according to reviewer’s suggestions

24. Line 308 – try “… 16 hours. Flies of the … exhibited …” 

Response: The text was revised according to reviewer’s suggestions

25. Line 309 – try “… activity. Therefore, we separated …” 

Response: The text was revised according to reviewer’s suggestions

26. Line 331 and 333 – replace “has been” with “was” – this is more direct and confident language

Response: The text was revised according to reviewer’s suggestions

27. Line 347-349 – try “Females face higher reproductive cost due to … events and this cost leads to decreased longevity.” 

Response: The text was revised according to reviewer’s suggestions

28. Line 351 – technically, lifespan should be “longer” because it related to time. The number might be “higher” but lifespan is longer. 

Response: The text was revised according to reviewer’s suggestions

29. Line 356 – technically, we say “our results are consistent with …”. Alignment is something different.

Response: The text was revised according to reviewer’s suggestions

30. Line 358 – do you mean “deteriorate” or “decorate”? 

Response: the word “decorate” replaced by the word “deteriorate”

31. Line 358-360 – try “… recorded. Young female … survival [18] but this strategy…” 

Response: The text was revised according to reviewer’s suggestions

32. Line 362 – I think you need to state that more energy use leads to death or shorter lifespan 

33. Response: Previous studies showed that increased flight activity and locomotion in Diptera, lead to shorter lifespan/ have a tradeoff in lifespan. In our study, there seems to be a similar trend of female medflies, depicted in the event- history graphs. However, the statistical analysis did not support this finding. Therefore, we prefer not to include this statement in our discussion.

34. Line 368 – try “… and higher frequency is positively …” 

Response: The text was revised according to reviewer’s suggestions

35. Line 375 – replace “has been” with “was” – replace “as well as” with “and” done

Response: The text was revised according to reviewer’s suggestions

36. Line 378-379 – try “Consistent with our results, daily …” 

Response: The text was revised according to reviewer’s suggestions

37. Line 389 – try the “consistent with” phrase rather than “agreement”.

Response: The text was revised according to reviewer’s suggestions

38. Line 389-392 – is very long – try two sentences.

Response: The text was revised according to reviewer’s suggestions

39. Line 396 – try “… day. Also, previous studies reported …” 

Response: The text was revised according to reviewer’s suggestions

40. Line 397 – try “…59]. Conversely, they exhibit …during later in the morning and midday.”

Response: The text was revised according to reviewer’s suggestions

41. Line 403 – use the “consistent with” phrase – not “agreement” 

Response: The text was revised according to reviewer’s suggestions

42. Line 411 – try “We found a …”. Technically, the results are a thing and it can’t do anything. Results can’t see or find or think. That is what the scientists, the humans, do. So use the “we” words. Results are something you create but results don’t do the thinking or showing. 

Response: The text was revised according to reviewer’s suggestions

43. Line 412-414. Say the important stuff first. Try “This system could be … aging process, provided that …” 

Response: The text was revised throughout according to reviewer’s suggestions

44. Line 417-420 – sentence too long. Try “… the trials … individuals. We successfully demonstrated several …” 

Response: The text was revised according to reviewer’s suggestions

45. References – some are abbreviated – some are in full. Pick the journal standard and stick to it

Response: The references were revised according to reviewer’s suggestion and the journal’s standards

Reviewer #3

A well-designed study investigating the lifelong locomotor activity of Medflies. Findings are providing valuable insights in the fecundity and activity patters of the flies during a period of many months (usually those studies are restricted in few 24 hrs periods for the purpose of evaluating quality of SIT flies. I would recommend this paper to be published, pending few minor corrections and suggestions that I submit for the consideration of the authors in the annotated PDF.

General comments:

1. It is known that social interactions and caging density affect the health of the flies, I would like to see some comment in discussion, since the experimental flies spent all their life under severely restricted solitary conditions. 

Response: Following reviewer’s suggestion, we included a paragraph in the discussion addressing the effect of social interaction on adult fruit flies.

2. The authors do mention the benefits of the modified LAM system for studying the activity patterns over very long periods of time. I would like to see in the discussion a brief comparison with other methods aiming to also access the organism’s physiological status over long period of time (lipids-proteins from subsamples, video recording?)

Response: Trying to connect activity patterns with physiological responses and longevity is an interesting subject that required a meticulously designed studies and adoption of elaborated protocol to see the dynamics of metabolic and other physiological processes. The current study approaches activity and life span from a demographic perspective and we prefer not to expand to physiological responses. The behavioral pattern of single individuals differentiates with aging. In our study we tried to investigate the connection between locomotory activity signature of flies with health and lifespan, which factors are strongly correlated with the physiological status. Therefore, such methods which made the correlation of lipids- proteins contents with aging (Nestel et al., 2004), cannot produce the type of data that we need. Other methods that have been applied to measure the behavior or activity during aging are the BMS (Behavioral Monitoring System) which cannot measure the detailed activity for every single minute and for long periods like the LAM25system. Generally, the other system cannot produce so detailed recordings regarding the locomotory activity for every minute during the 24h and additionally they cannot measure the activity during the night in the absence of light sources.

3. It would be interesting to access the effects of mating status, by monitoring mated vs virgin females (if not in this study, it would be worthy to mention it as basis for future experiments?)

Response: This in an excellent proposal for future studies regarding differences in locomotory activity between mated and virgin medflies. The cost of reproduction is strongly correlated with the aging process in medflies life history traits. Therefore, in future studies we agree that would be extremely interesting to investigate the differences between the locomotory activity patterns for virgin and mated flies during aging.

Specific comments:

1. Line 45 

Response: The text was revised according to reviewer’s suggestions

2. Line 53

Response: The text was revised according to reviewer’s suggestions

3. Line 71

Response: The text was revised according to reviewer’s suggestions

4. Line 77, 80, 81, 82

Response: The paragraph was revised thoroughly according to the reviewers’ suggestions

5. Line 84: need to clarify the mating status, as this is of importance for longevity

Response: The mating status of the flies was clarified revised according to reviewer’s suggestions

6. Line 88: maybe explain constrains?

Response: Some specific constrains added according to the reviewer’s suggestion

7. Line 132-134: This sentence belongs to the conclusions

Response: The text was revised according to reviewer’s suggestions

8. Line 142: light intensity?

Response: The light intensity was clarified according to reviewer’s suggestions

9. Line 155: dimensions of cotton disk? thickness?

Response: The cotton disk’s thickness and dimensions clarified

10. Line 250: clarify mating status

Response: The mating status was clarified

11. Line 339: how? it is not clear that females were mated, so only fecundity of unfertilized eggs is expected to be measured during those experiments

12. Response: The term ‘reproduction’ was replaced by the term ‘fecundity’ throughout the whole text. We also reported the mating status of the flies in the M&M section.

13. Line 344: only with reference to mated adult flies

Response: The text was revised for clarity

14. Line 363: reference to the restricted conditions needs to be presented here, since those flies did not had ability to fly in open space, or to develop social interactions that affect longevity.

15. Response: We thanks the reviewer for pointing this out. We have included a paragraph to address his concerns (see lines 377 - 386)

16. Line 418: Please note the number of individuals (males-females) and replications in M&M.

Response: The replicates were clarified in the M&M section

The data for the supplementary S2Table and S3Fig are given in: 

https://data.world/vasrod43/lamsystem-data-for-s2-table-and-s3-fig

---

## [Decision Letter · Decision Letter 1]

1 Jun 2022

Electronic recording of lifetime locomotory activity patterns of adult medflies

PONE-D-22-00246R1

Dear Dr. Papadopoulos,

We’re pleased to inform you that your manuscript has been judged scientifically suitable for publication and will be formally accepted for publication once it meets all outstanding technical requirements.

Kind regards,

Nicholas C. Manoukis

Academic Editor

PLOS ONE

Additional Editor Comments (optional):

Thank you for addressing all comments! Congratulations on completing a nice study.

Reviewers' comments:

Reviewer's Responses to Questions

**Comments to the Author**

1. If the authors have adequately addressed your comments raised in a previous round of review and you feel that this manuscript is now acceptable for publication, you may indicate that here to bypass the “Comments to the Author” section, enter your conflict of interest statement in the “Confidential to Editor” section, and submit your "Accept" recommendation.

Reviewer #1: All comments have been addressed

2. Is the manuscript technically sound, and do the data support the conclusions?

Reviewer #1: Yes

3. Has the statistical analysis been performed appropriately and rigorously? 

Reviewer #1: Yes

4. Have the authors made all data underlying the findings in their manuscript fully available?

Reviewer #1: Yes

5. Is the manuscript presented in an intelligible fashion and written in standard English?

Reviewer #1: Yes

6. Review Comments to the Author

Reviewer #1: The experiments described in this manuscript will prove useful for other pestiferous Tephritidae species and I therefor recommend this article for publication.

7. PLOS authors have the option to publish the peer review history of their article (what does this mean?). If published, this will include your full peer review and any attached files.

Reviewer #1: No

---

## [Editor Report · Acceptance letter]

7 Jun 2022

PONE-D-22-00246R1 

Electronic recording of lifetime locomotory activity patterns of adult medflies 

Dear Dr. Papadopoulos:

I'm pleased to inform you that your manuscript has been deemed suitable for publication in PLOS ONE. Congratulations! Your manuscript is now with our production department. 

Kind regards, 

on behalf of

Dr. Nicholas C. Manoukis 

Academic Editor

PLOS ONE